# Lack of Highly Pathogenic Avian Influenza H5N1 in the South Shetland Islands in Antarctica, Early 2023

**DOI:** 10.3390/ani14071008

**Published:** 2024-03-26

**Authors:** Gabriela Muñoz, Vanessa Mendieta, Mauricio Ulloa, Belén Agüero, Cristian G. Torres, Lucas Kruger, Victor Neira

**Affiliations:** 1Departamento de Medicina Preventiva Animal, Facultad de Ciencias Veterinarias y Pecuarias, Universidad de Chile, Santiago 8820808, Chile; munozbritogabriela@um.uchile.cl (G.M.); vmendietareis@gmail.com (V.M.); belen.aguero.a@gmail.com (B.A.); 2Programa de Doctorado en Ciencias Silvoagropecuarias y Veterinarias, Universidad de Chile, Santiago 8820808, Chile; 3Veterinary Histology and Pathology, Institute of Animal Health and Food Safety, Veterinary School, University of Las Palmas de Gran Canaria, 35001 Las Palmas de Gran Canaria, Spain; mulloa@sernapesca.cl; 4Servicio Nacional de Pesca y Acuicultura, Valparaíso 2340159, Chile; 5Departamento de Ciencias Clínicas, Facultad de Ciencias Veterinarias y Pecuarias, Universidad de Chile, Santiago 8820808, Chile; crtorres@uchile.cl; 6Instituto Antártico Chileno, Punta Arenas 6200000, Chile; lkruger@inach.cl; 7Millennium Institute of Biodiversity of Antarctic and Subantarctic Ecosystems (BASE), Ñuñoa 7750000, Chile

**Keywords:** highly pathogenic avian influenza, H5N1, Antarctica, lack of detection

## Abstract

**Simple Summary:**

A concerning highly pathogenic avian influenza caused widespread outbreaks among birds and wildlife at a global level. First identified in South America in October 2022 and linked to migratory birds, there’s growing concern about virus spillover during bird migrations between the poles. The Arctic tern’s migration from the Arctic to Antarctica raises concerns about its role in transmission. Even in Antarctica’s seemingly untouched environment, diverse wildlife is at risk from the new strain of highly pathogenic avian influenza. Surveillance efforts started in the South Shetland Islands in January 2023, focusing on areas where penguins, birds, and marine mammals normally gather. Following Antarctic Treaty guidelines, observations and sample collection were conducted, revealing no signs of the virus in the region during that time lapse. These findings not only contribute to confirming the absence of the virus in Antarctica during the specified period but also emphasize the necessity for sustained surveillance and monitoring to safeguard the Antarctic ecosystem.

**Abstract:**

In January 2023, an active surveillance initiative was undertaken in the South Shetland Islands, Antarctica, with the specific objective of ascertaining evidence for the presence of avian influenza, and specifically the highly pathogenic avian influenza virus subtype H5N1 (HPAIV H5N1). The investigation encompassed diverse locations, including Hanna Point (Livingston Island), Lions Rump (King George Island), and Base Escudero (King George Island), with targeted observations on marine mammals (southern elephant seals), flying birds (the kelp gull, snowy sheathbill and brown skua), and penguins (the chinstrap penguin and gentoo penguin). The study encompassed the examination of these sites for signs of mass mortality events possibly attributable to HPAIV H5N1, as well as sampling for influenza detection by means of real-time RT-PCR. Two hundred and seven (207) samples were collected, including 73 fecal samples obtained from the environment from marine mammals (predominantly feces of southern elephant seals), and 77 cloacal samples from penguins of the genus Pygoscelis (predominantly from the gentoo penguin). No evidence of mass mortality attributable to HPAIV H5N1 was observed, and all the collected samples tested negative for the presence of the virus, strongly suggesting the absence of the virus in the Antarctic territory during the specified period. This empirical evidence holds significant implications for both the ecological integrity of the region and the potential zoonotic threats, underscoring the importance of continued surveillance and monitoring in the Antarctic ecosystem.

## 1. Introduction

The highly pathogenic avian influenza subtype H5NX, clade 2.3.4.4b, has caused global epidemic outbreaks affecting both wildlife and poultry populations [1]. This virulent virus has permeated avian and mammalian species, instigating mass mortality events across several continents. In October 2022, South America registered the first detection of the virus in wild birds, attributed to migratory birds, with multiple introductions reported [2].

Given that several species of seabirds occurring in Antarctica have the capability of long-range movement, including transequatorial migrating species capable of traveling from the Arctic to Antarctica, great concerns arise about the potential dissemination of the virus during migratory events directly between polar circles. Two species stand out as potential contributors to such transmission, notably the arctic tern (*Sterna paradisaea*), well-known for undertaking one of the longest migratory journeys from the Arctic to Antarctica [3,4], and the south polar skua (*Catharacta maccormicki*) which spends the austral winter off the North Atlantic and Pacific oceans, and in tropical/subtropical waters off the North Indian Ocean [5,6]. Additionally, several other species that breed in Antarctica with shorter migratory routes but dispersive migration behaviors, when connected, might facilitate the introduction of the virus into the Antarctic continent, such as southern giant petrels (*Macronectes giganteus*) [7], southern fulmars (*Fulmarus glacialoides*), cape petrels (*Daption capense*) [8], and brown skuas (*Catharacta lonbergii*) [9,10]. Southern giant petrels have recently been reported as being able to move from the Antarctic Peninsula to the coast of Tierra del Fuego in Argentina and Chile and back in a few days [7].

So far, only low-pathogenic avian influenza strains like H11N2 [11,12] and H5N5 LPAIV [13] have been reported in Antartica, but the rich wildlife in the region remains highly susceptible to the new strains of H5NX clade 2.3.4.4b. Recent reports outside Antarctica have documented mass mortality events in penguins and marine mammals, the most abundant species in Antarctica. Humboldt penguins *(Spheniscus humboldti*), South American sea lions (*Otaria flavescens*) [14], southern elephant seals (*Mirounga leonina*) in South America, and African penguins (*Spheniscus demersus*) in Africa [15], along with other species in the Northern Hemisphere, have also been severely affected.

In view of the potential introduction of H5NX clade 2.3.4.4b, we initiated active surveillance in January 2023, four months after the introduction of HPAIV H5N1 in South America. Our surveillance efforts were strategically concentrated on selected locations within the Southern Shetland Islands, where penguins, flying birds, and marine mammals converge.

## 2. Materials and Methods

### 2.1. Locations

In January 2023, we performed active surveillance looking for mass mortality events in the Antarctic wildlife attributable to HPAIV H5N1. The surveyed locations included were selected based on the available logistics and preferred locations of penguins and marine mammals. Such locations included Hanna Point, Livingston Island, and Lions Rump near Low Head at King George Island, and the western shore of the Fildes Peninsula near Teniente R. Marsh Airport. Hanna Point (−62.6544, −60.6133), presents a high concentration of wildlife, such as nesting gentoo penguins (*Pygoscelis papua*), chinstrap penguins (*Pygoscelis antarcticus*), southern giant petrels (*Macronectes giganteus*), and the presence of southern elephant seals (*Mirounga leonina*). Lions Rump, (−62.1333, −58.1167), an Antarctic Specially Protected Area (ASPA No 151, permit 892/2022 INACH), presents nesting of Adelie (*Pygoscelis adeliae*), chinstrap, and gentoo penguins, and a large number of southern elephant seals and fur seals (*Arctocephalus australis*) along the beaches. The western shore of the Fildes Peninsula (−62.197778, −58.993611) is occupied by a colony of southern elephant seals; other marine mammals such as the fur seal, the leopard seal (*Hydrurga leptonyx*), and the Weddell seal (*Leptonychotes weddellii*) can also be observed. The surveillance team, made up of three individuals, fully adhered to Antarctic treaty guidelines when conducting the surveillance and monitoring of the selected locations, systematically observing and documenting the wildlife populations present. The locations are illustrated in Figure 1. The clinical observation and search for clinical signs of disease, such as respiratory, neurologic, or digestive syndromes, ref. [14] was performed by doctors of veterinary medicine.

The rationale behind the sampling strategy is related to the known susceptibility of penguins and elephant seals to HPAIV. We recognize that the primary risk of avian influenza reaching Antarctica most likely stems from migratory birds that regularly nest on the continent. We anticipate that upon the virus’s introduction, it would likely affect other at-risk species such as penguins and elephant seals. These species, which form substantial colonies, allow for more efficient evaluation and sample collection. This pattern is reflected in events observed in Peru, Argentina, and Chile in South America, where the introduction of the virus by migratory birds led to documented clinical observations of HPAIV penguins and marine mammals [16,17,18].

### 2.2. Sampling

A total of 207 samples, which consisted of 156 environmental samples and 51 cloacal samples, were collected to perform real-time reverse transcription-polymerase chain reaction (RT-PCR) targeting the M gene of the Influenza A virus. Specifically, at Hanna Point, Livingstone Island, a total of 80 samples were collected, including 53 from southern elephant seals, 13 from kelp gulls, and smaller numbers from the snowy sheathbill (*n* = 3), chinstrap penguins (*n* = 3), and gentoo penguins (*n* = 8).

During sampling in Lions Rump, King George Island, a total of 104 samples were collected. Among these, 68 originated from southern elephant seals, and there were varying counts from gentoo penguins (*n* = 33), Adelie penguins (*n* = 2), and chinstrap penguins (*n* = 1). On the west coast of the Fildes Peninsula, King George Island, 23 samples were collected, including from southern elephant seals and an unidentified mammal (six environmental samples), and a brown skua (Direct). In the case of marine mammals and flying birds, fresh feces from the environment were meticulously collected, while cloacal samples were collected from clinically healthy penguins. To gather samples, we carefully surrounded the colonies, seeking out fresh feces. Using sterile cotton swabs, samples were preserved in minimum essential medium supplemented with antibiotic and antimycotic (Gibco™ 15240062), and kept in frozen storage until processing. For penguins, the collection of samples involved the capture of the animals by a doctor of veterinary medicine using a bird net and restraint. Subsequently, a cloacal swab was taken and preserved in viral transport media for further analysis. Environmental sampling for marine mammals and flying birds was carried out based on convenience. In contrast, a minimum of 30 individual cloacal samples were systematically collected from locations inhabited by penguins to evaluate disease freedom. This number (30) allows for the detection of at least one positive sample given a prevalence in the location equal to or less than 10%.

### 2.3. Diagnostic Testing

The samples were frozen upon arrival at the Animal Virology Lab, Facultad de Ciencias Veterinarias y Pecuarias, Universidad de Chile. The total RNA was extracted using TRIzol^®^ LS Reagent (Invitrogen™, Carlsbad, CA, USA) following the manufacturer’s recommendations. The real-time reverse transcription-polymerase chain reaction (real-time RT-PCR) with TaqMan probe (IDT) was performed, amplifying a conserved region of the matrix gene [19,20]. This protocol allows one to detect the generic Influenza A virus, which include Avian Influenza and HPAIVs.

Samples with a cycle threshold (Ct) below 35 were considered positive, between 35 and 40 inconclusive, and repeated Ct over 40 were considered negative. If a positive sample was obtained, a real-time RT-PCR with TaqMan probes for H5 2.3.4.4b clade confirmation using VSL-USDA protocols could be attempted.

## 3. Results

### 3.1. No Clinical Signs Related to HPAIV

The surveillance was performed by exploring all the colonies in each selected location (Figure 1). In Hanna Point, three large groups of elephant seals, a nesting colony of kelp gulls (*Larus dominicanus*), a nesting colony of giant petrels, and a large colony of gentoo penguins were observed. Notably, no evidence of unexpected mortality events, such as deceased animals or carcasses, were observed. The observed animals displayed normal behavior, and there were no discernible signs of evident disease. In Lions Rump, a large group of elephant seals and colonies of Adelie and gentoo penguins were observed to exhibit normal behavior, and no fresh dead animals or carcasses were observed. Likewise, no signs of disease were observed on the western shore of the Fildes Peninsula, where a group of southern elephant seals in good physical shape was identified. The animals presented in good condition and no mortality was identified. Fur seals were not observed during the surveillance.

### 3.2. Lack of Detection by Real-Time RT PCR

For the total of the samples analyzed, all resulted in a negative outcome for Influenza A detection. In the initial testing, only three samples resulted as suspect with Ct values ranging from 36.5 to 37.8, all obtained from gentoo penguin chicks; however, they resulted negative during the confirmation round. The complete sampling is summarized in Table 1.

## 4. Discussion

The current outbreak of HPAIV H5N1, which began in 2022, has resulted in the deaths of a large number of seabirds across the Northern Hemisphere, Southern Africa, the Atlantic and Pacific Oceans, and throughout South America.

This study represents the first systematic effort to surveil the Influenza virus in Antarctica during 2023, focusing specifically on the South Shetland Islands near the Antarctic Peninsula in both avian and marine mammals. Fortunately, we neither observed clinical signs nor detected clinical or molecular evidence of the virus during surveillance in early 2023 at Lions Rump and Hanna Point, two distinctive areas for biodiversity in Antarctica inhabited by flying birds, penguins, and marine mammals, in any of the studied species. In a study primarily based on clinical observations in early 2023, no signs of the virus in the Antarctic and sub-Antarctic territories were reported either [21].

Migratory birds are pivotal in the global spread of HPAIV H5N1 [22]. The extensive infections documented in North America in 2022 indicate the potential for the virus to be transmitted over long distances, possibly even between polar regions. Although direct transmission over such vast distances is less common, there is substantial evidence of the virus’s spread from North America to Chile and Peru, and its further dissemination to remote areas like Patagonia [2,14,16,18].

Arctic terns, known for their pole-to-pole migrations, provide a direct transmission route for HPAIV H5N1 from the Northern Hemisphere [3]. These birds, vulnerable to the Influenza A virus, have been confirmed as carriers of HPAIV H5N1 from September to December 2022 in the Northern Hemisphere [23], and massive mortality events in colonies on wild birds have been linked to this strain [24,25]. Moreover, HPAIV H5N1 has been completely sequenced from Arctic terns collected in Alaska, Wales, Denmark, Germany, Netherlands, England, and Scotland between 2022 and 2023 (Supplementary Information). Historical records also show the presence of H5NX in Arctic terns, with the first sequenced virus being H5N3 collected in 1975 (Supplementary Information). The evidence suggests that Arctic terns are highly susceptible to the virus, which can cause significant mortality; however, some individuals may resist and migrate while carrying the virus. In contrast, the south polar skua, another species that migrates between hemispheres, may be susceptible to Influenza infection, but there is currently no evidence of this.

The circumstantial evidence indicates that there was most likely no effective transmission of HPAIV H5N1 clade 2.3.4.4b in Antarctica during the summer of 2022/2023 in the Southern Hemisphere. Nevertheless, the virus could reach the continent directly from the Northern Hemisphere, most likely via Arctic terns.

Another possibility is short-distance dissemination. The virus is already in South America and is more than likely to be introduced to Antarctica through local migrations, very probably during this current 2023–2024 season. Species at risk for the introduction of the virus include, but are not limited to, southern giant petrels, southern fulmars, Cape petrels, and brown skuas. Of course, after introduction, other species such as penguins and marine mammals should be affected, as mentioned before. It is important to note that in Chile more than 50 avian species have tested positive [26].

Biologically speaking, the difference between HPAIV and low-pathogenic avian influenza virus (LPAIV) lies in the fact that HPAIV is a systemic infection, while LPAI remains localized in the respiratory and intestinal tract. In this study, cloacal swabs, a recommended choice for waterfowl, were used. This differs from gallinaceous poultry, for which oropharyngeal swabs are recommended; however, it is considered optimal to collect swabs from both anatomical areas [27]. In this study, the collection of oropharyngeal samples was not chosen due to the invasive nature of this technique, particularly in wild bird species. Instead, cloacal swabs and environmental fecal samples, representing a less intrusive option for sampling, were utilized.

The HPAIV H5N1 clade 2.3.4.4b is approaching Antarctica; it reached Tierra del Fuego, Chile, in April 2023 [26] (https://websag.azurewebsites.net/ia, accessed on 17 January 2024). Then, it was later confirmed in the Sub-Antarctic region, specifically in South Georgia [28], where mortalities in brown skuas, kelp gulls, and southern elephant seals were observed. The sequenced viruses from these events are linked to those from Chilean cases [28]. In the Falkland Islands, infections were confirmed in southern fulmars and black-browed albatrosses (*Thalassarche melanophris*). Due to the fact that all these species can reach Antarctica, the potential for the virus to spread is significant.

It is crucial to note that migratory wild birds pose a significant risk upon their next arrival, as opposed to the Procellariiformes, which constantly move throughout the region. Southern giant petrels, for example, can travel from the Antarctic Peninsula to the coast of Tierra del Fuego in Chile and Argentina and be back within a few days [7], even while tending to active nests. Due to their scavenging behavior, giant petrels are frequent visitors to penguin colonies and seal haul-out sites in Antarctica [29]. Immature southern giant petrels are often observed in the Magellan Strait, close to shore, and are known to interact with a variety of coastal species. There is no information on how these immature birds interact with conspecific adults year-round, but given the high mobility of the immature individuals, it is likely that some level of contact exists. Black-browed albatrosses also undertake foraging trips to the Antarctic Peninsula, and since at least one positive case has been reported in an albatross [30], they become another potential vector.

Finally, while elephant seal fatalities on Livingston Island are suspected, no positive detection of the virus has been confirmed, including the case of three dead elephant seals sampled by nasal and rectal swabs in Moltke Harbour [28]. However, a recent outbreak of HPAIV H5N1 in southern elephant seal pups was verified in Peninsula Valdés and adjacent areas of Argentina [18]. Additionally, the presence of leopard seals, which prey on birds such as penguins, introduces a new variable in the epidemiology of the disease, in that some species may contract the virus via the digestive route [14]. However, mammal-to-mammal transmission has recently been suggested [31].

Several iconic species could be affected by the HPAIV H5N1. Low-pathogenic avian influenza virus (LPAIV) subtype H11N2 is endemic among penguin populations, with low prevalence using direct molecular methods [11,12]. Reassortant strains with segments from South America [32], North America, and Eurasia [13] have also been detected. An LPAI H5N5 strain was identified in Antarctica in 2015 [12,13], suggesting its spread; however, it has not been detected ever since, indicating that non-adapted strains may die out. There is a possibility that penguins exhibit varying levels of susceptibility to HPAIV H5N1. Circumstantial evidence suggests that the Humboldt penguin may be more susceptible than the Magellanic penguin, due to the high impact in mortality rates of Humboldt penguins recorded in Chile (20%) during 2023 compared to Magellanic penguins (3%), where most of the individuals have died from bycatch and so far no registered case from HPAI H5N1 virus has been registered, if we consider that both species share the same habitat along the south-eastern Pacific coast of Chile [33], a hypothesis that requires further confirmation.

Regarding marine mammals, the susceptibility of southern elephant seals has already been confirmed [18]. Regarding fur seals, the only confirmation of the virus has recently been reported in Uruguay [31]. From the above, it is evident that Antarctic pinnipeds are susceptible species that must necessarily be considered in new studies of Antarctica.

## 5. Conclusions

This study was conducted in January 2023 across three diverse wildlife locations in the South Shetland Islands. We relied on clinical observations and molecular detection to investigate the presence of HPAIV H5N1. However, we found no evidence of the virus. This research, along with other studies, strongly suggests that the virus did not reach the Antarctic continent in the last austral summer of 2022/2023. Although migratory birds, such as Arctic terns, were identified as potential vectors, no clinical signs of the virus were found during surveillance activities. It is important to highlight that the current findings may be influenced and limited by the sample size and specific locations covered during the study. To enhance the reliability of the results pertaining to the presence or absence of the virus, is necessary to increase the number of samples from various locations in a new study in the current season.

Importantly, endemic LPAIV viruses exhibit a different infection dynamic compared to HPAIV H5N5. HPAIV outbreaks are characterized by high morbidity and mortality, with the virus being detected in most of the affected and deceased animals. In this sense, expert clinical observations and molecular diagnostics are complementary strategies to determine the presence of this virus. Conversely, the previously detected endemic LPAIV H11N2 virus occurs with low prevalence in the absence of clinical signs. Hurt et al. in 2016 found one positive out of 295 samples in 2014, and one out of 493 in 2015 [12].

This research emphasizes the potential risk of introducing the virus from the Northern Hemisphere and currently from South America through local and long-distance migratory birds, highlighting the importance of ongoing surveillance efforts. As the virus poses a potential threat to nearby regions such as Tierra del Fuego and South Georgia, along with the possibility of future spread to Antarctica, it is imperative to expand the scope of surveillance activities.

The arrival of the virus in nearby sub-Antarctic regions and its potential spread underscores the need to understand and monitor transmission patterns in this unique environment. Given the susceptibility of various species, including Antarctic pinnipeds, continuous research is crucial to safeguard this delicate ecosystem. Having a robust and more comprehensive dataset through increased sampling is indispensable to the process of making informed decisions and developing effective strategies to prevent and manage the potential introduction of the HPAIV H5N1 to Antarctica.

## Figures and Tables

**Figure 1 animals-14-01008-f001:**
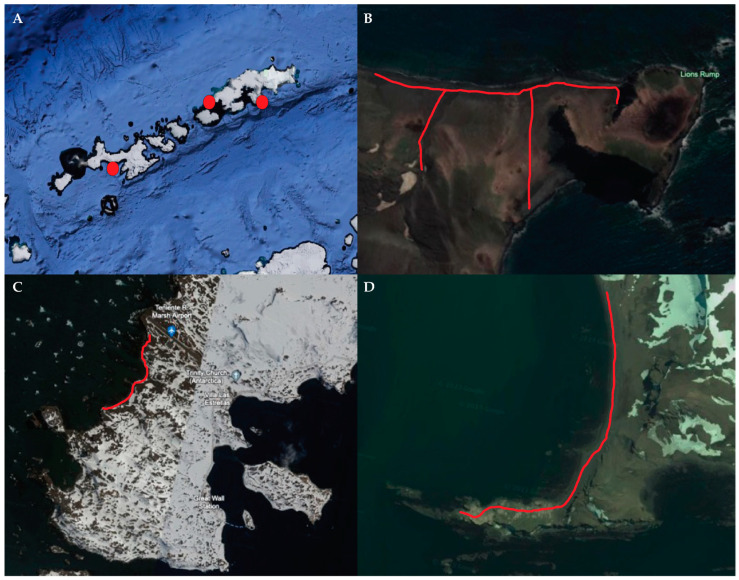
Image of the surveyed locations in the South Shetland Islands, Antarctica, captured from Google Earth. (**A**) South Shetland Islands with red dots that indicate the sampling locations. (**B**) Lions Rump in King George Island. (**C**) Fildes Peninsula in King George Island. (**D**) Hanna Point in Livingstone Island. The red line indicates the trekking route during the surveillance and monitoring activities. Captures are not to scale.

**Table 1 animals-14-01008-t001:** Overall sample collection in surveillance and monitoring activities. All samples tested negative for influenza identification.

Locations	Especies	Environmental	Direct	Total
Hanna Point, Livingstone Island	Southern elephant seal	53	0	53
	Kelp gull	13	0	13
	Snowy sheathbill	0	3	3
	Chinstrap penguin	0	3	3
	Gentoo penguin	0	8	8
Lions Rump, King George Island	Southern elephant seal	68	0	68
	Gentoo penguin	0	33	33
	Adelie penguin	0	1	1
	Chinstrap penguin	0	2	2
Fildes Peninsula western coast, King George Island	Southern elephant seal	16	0	16
	Unidentified mammal	6	0	6
	Brown skua	0	1	1
	Total	156	51	207

## Data Availability

Data are contained within the article and Appendix A.

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
