# Peer review of "Lack of Highly Pathogenic Avian Influenza H5N1 in the South Shetland Islands in Antarctica, Early 2023"

_animals, 2024, doi:10.3390/ani14071008_

Round 1

Reviewer 1 Report

Comments and Suggestions for Authors

Gabriela and coauthors performed a surveillance of H5N1 highly pathogenic avian influenza virus in Southern Shetland Islands in Antarctica in early 2023. They tested a total of 207 samples including environmental and cloacal samples directly from animals. The testing method is real-time RT PCR for influenza M gene. All the samples are negative. The result shows that based on this surveillance, no influenza A virus is detected. However, if the authors conclude that there is no H5N1 Highly pathogenic avian influenza virus, more sampling is needed. In this area, H5N5 and H11N2 had been reported, but this study did not detect any. Therefore, the results in this study can not support the conclusion.

Author Response

Reviewer 1:

Gabriela and coauthors performed a surveillance of H5N1 highly pathogenic avian influenza virus in Southern Shetland Islands in Antarctica in early 2023. They tested a total of 207 samples including environmental and cloacal samples directly from animals. The testing method is real-time RT PCR for influenza M gene. All the samples are negative. The result shows that based on this surveillance, no influenza A virus is detected.

  • However, if the authors conclude that there is no H5N1 Highly pathogenic avian influenza virus, more sampling is needed.

Response: Accepted. We have incorporated modifications into the main text and conclusions to address the reviewer's comments. We acknowledge the reviewer's point that the number of samples is a critical factor in determining the absence of HPAIV H5N1. It is, however, important to note that both clinical surveillance and sampling have led us to conclude that HPAIV was neither observed clinically nor identified through molecular diagnostics. HPAIV has a different viral dynamic compared to LPAIV see next comment.  Furthermore, we recognize that increased sampling would enhance the accuracy of our conclusions. Nonetheless, even with a relatively small number of samples, this represents a strength compared to other studies that rely solely on clinical observations not conducted by veterinarians.

Now incorporated 292-309

This study was conducted in January 2023 across three diverse wildlife locations in the Southern Shetland Islands, we relied on clinical observations and molecular detection to investigate the presence of HPAIV H5N1. However, we found no evidence of the virus. This research, along with other studies, strongly suggests that the virus did not reach the Antarctic continent in the last austral summer of 2022/2023. Although migratory birds, such as Arctic terns, were identified as potential vectors, no clinical signs of the virus were found during surveillance activities. It is important to highlight that the current findings may be influenced and limited by the sample size and specific locations covered during the study. To enhance the reliability of the results on the presence or absence of the virus, is necessary to increase the number of samples from various locations in a new study in the current season.

Importantly, endemic LPAIV viruses exhibit a different infection dynamic compared to HPAIV H5N5. HPAIV outbreaks are characterized by high morbidity and mortality, with the virus being detected in most of the affected and deceased animals. In this sense, expert clinical observations and molecular diagnostics are complementary strategies to determine the presence of this virus. Conversely, the LPAIV H11N2 endemic previously detected occurs with low prevalence in the absence of clinical signs. Hurt et al. (2016) found one positive out of 295 samples in 2014, and one out of 493 in 2015.

  • In this area, H5N5 and H11N2 had been reported, but this study did not detect any.

Response: Accepted. We have incorporated the suggested changes. It is important to note that endemic LPAIV viruses exhibit a different infection dynamic compared to HPAIV H5N5. HPAIV outbreaks are characterized by high morbidity and mortality, with the virus being detected in most of the affected and deceased animals. Therefore, clinical observation by experts and molecular diagnostics are complementary methods to determine the presence of this virus. Conversely, the LPAIV previously detected in certain regions of Antarctica shows a distinct viral dynamic. The H11N2 virus, which is endemic in Antarctic penguins, occurs in the absence of clinical signs. This endemic virus is found in healthy populations, with only a low proportion detected by molecular diagnostics. For instance, Ogrzewalska et al. (2022) detected AIV (H11N2) in just one of the eight locations sampled in the Shetland Islands, and Hurt et al. (2016) found one out of 295 samples to be positive by PCR in 2014, and one out of 493 in 2015. We have added text to clarify these differences, emphasizing that the study focuses on HPAIV H5N1 rather than other endemic LPAIV viruses.

Now 303-310

Importantly, endemic LPAIV viruses exhibit a different infection dynamic compared to HPAIV H5N5. HPAIV outbreaks are characterized by high morbidity and mortality, with the virus being detected in most of the affected and deceased animals. In this sense, expert clinical observations and molecular diagnostics are complementary strategies to determine the presence of this virus. Conversely, the LPAIV H11N2 endemic previously detected occurs with low prevalence in the absence of clinical signs. Hurt et al. (2016) found one positive out of 295 samples in 2014, and one out of 493 in 2015.

  • Therefore, the results in this study can not support the conclusion.

Response: Partially accepted. We modified the conclusions. However, we considered that both clinical observations and molecular detection support our conclusion which is defined in a specific area and temporal frame.

Now 292 297

This study was conducted in January 2023 across three diverse wildlife locations in the Southern Shetland Islands, we relied on clinical observations and molecular detection to investigate the presence of HPAIV H5N1. However, we found no evidence of the virus. This research, along with other studies, strongly suggests that the virus did not reach the Antarctic continent in the last austral summer of 2022/2023. 

Reviewer 2 Report

Comments and Suggestions for Authors

Studies of this type have been conducted globally, however, this is understood as this type of study needs to be continuously conducted for surveillance purposes

The conclusion made is too strong compared to the evidence provided esp. the small size samples for the authors to come to this conclusion. i.e  confirming the absence of the virus in Antarctica during the specified period. Only environmental samples and about 50 cloacal samples have been collected. Besides small sample size, should consider of  tissue tropism of HPAIV for the upper respiratory tract than for the gastrointestinal tract.   It is also difficult to conclude that HPAIV is rare among the healthy   populations of wild migratory birds in your study. 

Should add reference: No HPAIV was detected in large surveillance studies conducted in the U.S., Europe, Alaska, Africa, Iran,and Mongolia.

Provide limitations for the study to be considered for the negative detection of the viruses, and better conclusions can be made from this study 

Comments on the Quality of English Language

Some sentences need to be rephrased 

eg. The virus is already in SA (line 200) to the virus is reported to be present or prevalent in SA

eg `we observed neither clinical evidence nor molecular detection of the virus' please add in which animals?

Author Response

Reviewer 2:

Studies of this type have been conducted globally, however, this is understood as this type of study needs to be continuously conducted for surveillance purposes.

  • The conclusion made is too strong compared to the evidence provided esp. the small size samples for the authors to come to this conclusion. i.e  confirming the absence of the virus in Antarctica during the specified period.

Response: Accepted. We have incorporated modifications into the main text and conclusions to address the reviewer's comments. We acknowledge the reviewer's point that the number of samples is a critical factor in determining the absence of HPAIV H5N1. It is, however, important to note that both clinical surveillance and sampling have led us to conclude that HPAIV was neither observed clinically nor identified through molecular diagnostics. HPAIV has a different viral dynamic compared to LPAIV see next comment.  Furthermore, we recognize that increased sampling would enhance the accuracy of our conclusions. Nonetheless, even with a relatively small number of samples, this represents a strength compared to other studies that rely solely on clinical observations not conducted by veterinarians.

Now incorporated 292-309

This study was conducted in January 2023 across three diverse wildlife locations in the Southern Shetland Islands, we relied on clinical observations and molecular detection to investigate the presence of HPAIV H5N1. However, we found no evidence of the virus. This research, along with other studies, strongly suggests that the virus did not reach the Antarctic continent in the last austral summer of 2022/2023. Although migratory birds, such as Arctic terns, were identified as potential vectors, no clinical signs of the virus were found during surveillance activities. It is important to highlight that the current findings may be influenced and limited by the sample size and specific locations covered during the study. To enhance the reliability of the results on the presence or absence of the virus, is necessary to increase the number of samples from various locations in a new study in the current season.

Importantly, endemic LPAIV viruses exhibit a different infection dynamic compared to HPAIV H5N5. HPAIV outbreaks are characterized by high morbidity and mortality, with the virus being detected in most of the affected and deceased animals. In this sense, expert clinical observations and molecular diagnostics are complementary strategies to determine the presence of this virus. Conversely, the LPAIV H11N2 endemic previously detected occurs with low prevalence in the absence of clinical signs. Hurt et al. (2016) found one positive out of 295 samples in 2014, and one out of 493 in 2015.

  • Only environmental samples and about 50 cloacal samples have been collected. Besides small sample size, should consider of  tissue tropism of HPAIV for the upper respiratory tract than for the gastrointestinal tract.

Response: Accepted. The number of samples was incorporated in the previous comment. About the viral tropism, we incorporated a paragraph in the discussion.

Now 233-240

Biologically speaking, the difference between HPAIV and low pathogenic avian influenza virus (LPAIV) lies in that HPAIV is a systemic infection, while LPAI remains localized in the respiratory and intestinal tract. In this study, cloacal swabs were used, a recommended choice for waterfowl. This differs from gallinaceous poultry, for which oropharyngeal swabs are recommended; however, it is considered optimal to collect swabs from both anatomical areas [30]. In this study, the collection of oropharyngeal samples was not chosen due to the invasive nature of this technique, particularly in wild bird species. Instead, cloacal swabs and environmental fecal samples were utilized, representing a less intrusive option for sampling.

  • It is also difficult to conclude that HPAIV is rare among the healthy  populations of wild migratory birds in your study. 

Response: Accepted. The reviewer pointed this out. To better understand the rationale of the study we incorporated a new paragraph in the main text. 

Now 110-118.

The rationale behind the sampling strategy is related to the known susceptibility of penguins and elephant seals to HPAIV. We recognize that the primary risk of avian influenza reaching Antarctica most likely stems from migratory birds that regularly nest on the continent. We anticipate that upon the virus's introduction, it would likely affect other at-risk species such as penguins and elephant seals. These species, which form substantial colonies, allow for more efficient evaluation and sample collection. This pattern is reflected in events observed in Peru, Argentina, and Chile in South America, where the introduction of the virus by migratory birds led to documented clinical observations of HPAIV penguins and marine mammals [16–18]

  • Should add reference: No HPAIV was detected in large surveillance studies conducted in the U.S., Europe, Alaska, Africa, Iran, and Mongolia.

Response: We don’t understand the comment. On the contrary, there is evidence of the virus in all mentioned places.

Provide limitations for the study to be considered for the negative detection of the viruses, and better conclusions can be made from this study 

Response: Accepted. We incorporated the limitations, and the conclusion was rewritten.

Now incorporated 293-323

This study was conducted in January 2023 across three diverse wildlife locations in the Southern Shetland Islands, we relied on clinical observations and molecular detection to investigate the presence of HPAIV H5N1. However, we found no evidence of the virus. This research, along with other studies, strongly suggests that the virus did not reach the Antarctic continent in the last austral summer of 2022/2023. Although migratory birds, such as Arctic terns, were identified as potential vectors, no clinical signs of the virus were found during surveillance activities. It is important to highlight that the current findings may be influenced and limited by the sample size and specific locations covered during the study. To enhance the reliability of the results on the presence or absence of the virus, is necessary to increase the number of samples from various locations in a new study in the current season.

Importantly, endemic LPAIV viruses exhibit a different infection dynamic compared to HPAIV H5N5. HPAIV outbreaks are characterized by high morbidity and mortality, with the virus being detected in most of the affected and deceased animals. In this sense, expert clinical observations and molecular diagnostics are complementary strategies to determine the presence of this virus. Conversely, the LPAIV H11N2 endemic previously detected occurs with low prevalence in the absence of clinical signs. Hurt et al. (2016) found one positive out of 295 samples in 2014, and one out of 493 in 2015.

This research emphasizes the potential risk of introducing the virus from the Northern hemisphere and currently from South America by local and long-distance migratory birds, highlighting the importance of ongoing surveillance efforts. As the virus poses a potential threat to nearby regions such as Tierra del Fuego and South Georgia, along with the possibility of future spread to Antarctica, it is imperative to expand the scope of surveillance activities.

The arrival of the virus in nearby sub-Antarctic regions and its potential spread underscores the need to understand and monitor transmission patterns in this unique environment. Given the susceptibility of various species, including Antarctic pinnipeds, continuous research is crucial to safeguard this delicate ecosystem. Having a robust and more comprehensive dataset through increased sampling is indispensable for making informed decisions and developing effective strategies will be achieved to prevent and manage the potential introduction of the HPAIV H5N1 to Antarctica.

Some sentences need to be rephrased 

  1. The virus is already in SA (line 200) to the virus is reported to be present or prevalent in SA.

Response: Accepted. The paragraph was amended.

eg `we observed neither clinical evidence nor molecular detection of the virus' please add in which animals?

Response: Accepted.

Now 196-199. Fortunately, we observed neither clinical signs nor detected clinical or molecular evidence of the virus during early 2023 surveillance at Lions Rump and Hanna Point, two distinctive areas for biodiverse in Antarctica inhabited by flying birds, penguins, and marine mammals, and in none of the studied species.

Reviewer 3 Report

Comments and Suggestions for Authors

I have thoroughly read the manuscript named "Lack of Highly Pathogenic Avian Influenza H5N1 in Southern 2 Shetland Islands in Antarctica, early 2023". I found the subject interesting and adequate, the topic relevant and efforts significant in the sense of preservation of wildlife due to global climate change. Before publication, there are some issues to be addressed. First comment is the lack of exact sample numbers in material and methods section. Even though listed afterwards in the results, this need to be clearly stated before. In the detection of avian influenza two protocols targeting general influenza A and H5N1 were used. For the three samples with Ct values in the zone of suspicious results, which protocol was used to decide if they are negative? From the text, it reads that they were tested with H5 protocol and then ruled out as negative. If this is case still those three results should be discussed as inconclusive for other influenza A viruses. If this is not a case, please make it clear in the text. The role of marine mammals in the epidemiology of avian influenza is mentioned in a minor way in the text. Nevertheless, the substantial number of samples is collected from marine mammals. I suggest including more data on species distribution and significance for the ecosystem. Also, unidentified mammal in the table in result section should be discussed for the general readers in introduction. Finally, please describe catching of the birds and state the existing ethical statement in the text. The other minor issues are listed point by point in the text below:

Line 21. Please replace the term "there is worry" with more adequate one.

Line 35. I suggest listing wildlife species for part of the sentence " marine mammals, Antarctic birds, and Pygoscelis penguins" (such as....) since terms are too wide.

Line 39. Please define environmental samples. I also suggest dividing in categories with exact numbers of samples. 

Line 47. Keywords- I suggest replacing avian influenza or HPAI term with H5N1 to be more accurate.

Line 91. I suggest writing in decimal degrees.

Line 95. Please write Latin name of the species in the first occurrence in the text

Line 98. Please define other marine mammals

Line 103. Doctors of veterinary medicine

Line 108. Please define exactly how many samples of which kind was collected. Please write how many on which locations. 

Line 110. Please state which viral transportation medium was used.

Line 116. Please explain on which basis (colony sizes) was 30 samples threshold calculated. I strongly believe that this data should be well explained the text.

Line 120. I think total RNA was isolated by protocol, please state it exactly. 

Line 123. Please state chemical used to perform standardized protocol or make clear that the same were used. 

Line 142. Please avoid interesting since this is a subjective statement. 

Line 153. this sentence belongs to material and methods section in part. 

Line 155. Please replace "in the first round" with more adequate expression since there are two different protocols applied with different targets.

Line 162-167. This paragraph is a repetition. 

Line 181. Please state the reference.

Line 195. I suggest referring to own results in the discussion session nevertheless the results are negative and on the limited number of samples. 

Line 211. Please state the Latin species name for all species. Please uniform this issue throughout the text. 

Line 233-245. Please discuss in-depth manner the susceptibilities mentioned in this paragraph. 

Comments on the Quality of English Language

Minor editing of English language is needed.

Author Response

Reviewer 3:

I have thoroughly read the manuscript named "Lack of Highly Pathogenic Avian Influenza H5N1 in Southern 2 Shetland Islands in Antarctica, early 2023". I found the subject interesting and adequate, the topic relevant and efforts significant in the sense of preservation of wildlife due to global climate change. Before publication, there are some issues to be addressed. First comment is the lack of exact sample numbers in material and methods section. Even though listed afterwards in the results, this need to be clearly stated before. In the detection of avian influenza two protocols targeting general influenza A and H5N1 were used. For the three samples with Ct values in the zone of suspicious results, which protocol was used to decide if they are negative? From the text, it reads that they were tested with H5 protocol and then ruled out as negative. If this is case still those three results should be discussed as inconclusive for other influenza A viruses. If this is not a case, please make it clear in the text. The role of marine mammals in the epidemiology of avian influenza is mentioned in a minor way in the text. Nevertheless, the substantial number of samples is collected from marine mammals. I suggest including more data on species distribution and significance for the ecosystem. Also, unidentified mammal in the table in result section should be discussed for the general readers in introduction. Finally, please describe catching of the birds and state the existing ethical statement in the text. The other minor issues are listed point by point in the text below:

 Line 21. Please replace the term "there is worry" with more adequate one.

Response: Accepted; the word was amended. Now: concern.

 Line 35. I suggest listing wildlife species for part of the sentence " marine mammals, Antarctic birds, and Pygoscelis penguins" (such as....) since terms are too wide.

R: Accepted, the exact information was incorporated. Now: marine mammals (Southern Elephant Seals), flying birds (Kelp Gull, Snowy Sheathbill, and Brown Skua), and penguins (Chinstrap Penguin and Gentoo Penguin).

 Line 39. Please define environmental samples. I also suggest dividing in categories with exact numbers of samples. 

R: Accepted, the exact information for the abstract has been clarified.

Now lines 40-42

Two hundred and seven (207) samples were collected, including 73 fecal samples obtained from the environment from marine mammals (predominantly feces of Southern Elephant Seals), and 77 cloacal samples from penguins of the genus Pygoscelis (predominantly from Gentoo Penguin).

Line 47. Keywords- I suggest replacing avian influenza or HPAI term with H5N1 to be more accurate.

Response: Accepted. The word was amended in keywords.

Line 91. I suggest writing in decimal degrees.

Response: Accepted. All locations were changed to degrees.

Line 95. Please write Latin name of the species in the first occurrence in the text

Response: Accepted. The Latin name of the species in the first occurrence in the text was incorporated.

Line 98. Please define other marine mammals

Response: Accepted. Now lines 101-104: other marine mammals such as Fur Seal, Leopard Seal (Hydrurga leptonyx) and Weddell Seal (Leptonychotes weddellii) can be also observed. 17) Line 103.

Doctors of veterinary medicine

Response: Accepted and changed

Line 108. Please define exactly how many samples of which kind was collected. Please write how many on which locations. 

Response: Accepted, the detail of the number and origin of the samples was specified. Also is found in the table 1.

Line 110. Please state which viral transportation medium was used.

Response: Accepted. Now lines 36-37: preserved in minimum essential medium (Corning) supplemented with antibiotic and antimycotic (Gibco™ 15240062), and storage frozen until processing

Line 116. Please explain on which basis (colony sizes) was 30 samples threshold calculated. I strongly believe that this data should be well explained the text.

Response: Accepted. The sentence was rephrased to improve.

Now in lines 141-144

In contrast, a minimum of 30 individual cloacal samples were systematically collected from locations inhabited by penguins to evaluate disease freedom. This number (30) allows to detection of at least one positive sample given a prevalence in the location equal to or less than 10%.

Line 120. I think total RNA was isolated by protocol, please state it exactly. 

Response: Accepted. The word was amended.

 Line 123. Please state chemical used to perform standardized protocol or make clear that the same were used. 

Response: Accepted, the information was clarified.

Now 151-152.

The real-time reverse transcription-polymerase chain reaction (real-time RT-PCR) with TaqMan probe (IDT) was performed, amplifying a conserved region of the matrix gene [19,20].

Line 142. Please avoid interesting since this is a subjective statement. 

Response: Accepted. The word was removed.

Line 153. this sentence belongs to material and methods section in part. 

Response: Accepted, the paragraph was moved and replaced.

 Line 155. Please replace "in the first round" with more adequate expression since there are two different protocols applied with different targets.

Response: Accepted, the text was modified to “confirmation”.

Line 162-167. This paragraph is a repetition. 

Response: Accepted, redundant information was removed.

Line 181. Please state the reference.

Response: Accepted. The references were incorporated.

Line 195. I suggest referring to own results in the discussion session nevertheless the results are negative and on the limited number of samples.

Response: Accepted. The discussion and conclusions were rewritten, and the limitations were incorporated.

Now lines 293-323

This study was conducted in January 2023 across three diverse wildlife locations in the Southern Shetland Islands, we relied on clinical observations and molecular detection to investigate the presence of HPAIV H5N1. However, we found no evidence of the virus. This research, along with other studies, strongly suggests that the virus did not reach the Antarctic continent in the last austral summer of 2022/2023. Although migratory birds, such as Arctic terns, were identified as potential vectors, no clinical signs of the virus were found during surveillance activities. It is important to highlight that the current findings may be influenced and limited by the sample size and specific locations covered during the study. To enhance the reliability of the results on the presence or absence of the virus, is necessary to increase the number of samples from various locations in a new study in the current season.

Importantly, endemic LPAIV viruses exhibit a different infection dynamic compared to HPAIV H5N5. HPAIV outbreaks are characterized by high morbidity and mortality, with the virus being detected in most of the affected and deceased animals. In this sense, expert clinical observations and molecular diagnostics are complementary strategies to determine the presence of this virus. Conversely, the LPAIV H11N2 endemic previously detected occurs with low prevalence in the absence of clinical signs. Hurt et al. (2016) found one positive out of 295 samples in 2014, and one out of 493 in 2015.

This research emphasizes the potential risk of introducing the virus from the Northern hemisphere and currently from South America by local and long-distance migratory birds, highlighting the importance of ongoing surveillance efforts. As the virus poses a potential threat to nearby regions such as Tierra del Fuego and South Georgia, along with the possibility of future spread to Antarctica, it is imperative to expand the scope of surveillance activities.

The arrival of the virus in nearby sub-Antarctic regions and its potential spread underscores the need to understand and monitor transmission patterns in this unique environment. Given the susceptibility of various species, including Antarctic pinnipeds, continuous research is crucial to safeguard this delicate ecosystem. Having a robust and more comprehensive dataset through increased sampling is indispensable for making informed decisions and developing effective strategies will be achieved to prevent and manage the potential introduction of the HPAIV H5N1 to Antarctica.

Line 211. Please state the Latin species name for all species. Please uniform this issue throughout the text. 

Response: Accepted. We indicated all scientific names.

Line 233-245. Please discuss in-depth manner the susceptibilities mentioned in this paragraph. 

Response: Accepted. All the species mentioned have been reported as positive for HPAIV H5N1. We acknowledge the reviewer's observation regarding our mention of the potential differences in susceptibility between Magellanic and Humboldt penguins. We have expanded our discussion and included additional data about this aspect.

Now lines 278-284

Circumstantial evidence suggests that the Humboldt Penguin may be more susceptible than the Magellanic Penguin, due to the high impact in mortality rates of Humboldt Penguins recorded in Chile (20%) during 2023 compared to Magellanic Penguins (3%) where most of the individuals have died from bycatch and so far no registered case from HPAI H5N1 virus has been registered, if we consider that both species share the same habitat along the Southern Easter Pacific Ocean of Chile [36], a hypothesis that requires further confirmation.

Reviewer 4 Report

Comments and Suggestions for Authors

The paper deals with an issue that is very timely, and it should be emphasized that any information that relates to the spread of avian influenza, in this very dangerous variant, is important and necessary, for decision-making, both environmental locally and globally. It should be borne in mind that bird flu in Antarctica can do incredible damage, and therefore the issue taken up by the authors is important.

However, I have a few comments and doubts, which concern both the form in which the authors wrote this paper and the formal/methodical side. Let me start with the latter.

First, all sources state that the highest risk of bird flu transmission to Antarctica, might be connected with migration of five species which nest in Antarctica regularly (i.e., kelp gull (Larus dominicanus), Antarctic tern (Sterna vittata), the southern giant petrel (Macronectes giganteus), the south polar skua (Stercorarius maccormicki)  and the brown skua (Stercorarius antarcticus)). Only 14 of 207 samples were taken from the above mentioned species. If this is the case, it cannot be concluded that there was no bird flu in Antarctica in early 2023. So the title of the paper is confusing and should be corrected accordingly to the work done.

Second, the choice of locations was not clearly described. Why exactly these locations were chosen. Besides, as far as Lions Rump is concerned, it is an ASPA and sampling on it, requires the approval of the ethics committee. In the case of Chile, such approval is given by INACH, whether the team had such approval. If it did have it, why did it not take samples from the snowy sheathbill (Chionis albus) or brown skua (Stercorarius antarcticus), which nest there. It seems, that the sampling collection was done in haste and not necessarily in a thoughtful manner.  This needs clarifications. And the permit number for entrance to ASPA 151 should be stated clearly in the paper.

Third, the species names of animals should be standardized in accordance with accepted rules. This includes both English and Latin names. Sometimes authors wrote both words with a capital letter, sometimes whole guttural names, such as Sothern Giant Petrel, and sometimes short Giant Petrel. Both species of Giant Petrels are found in South Shetland, so this can be misleading. Table 1 is also a good example, in which one of the penguin spp. was written as: Gentoo penguin or Gentoo Penguin. Should be corrected accordingly.

I cannot find the supplementary files. This should be corrected. Generally, the paper is not well-organized and not well-written and should be rewritten. I encourage authors to present some graphics recording to your data. For example, graph related to the number of samples from different localizations.

Comments on the Quality of English Language

Third, the species names of animals should be standardized in accordance with accepted rules. This includes both English and Latin names. Sometimes authors write both words with a capital letter, sometimes whole guttural names, such as Sothern Giant Petrel, and sometimes short Giant Petrel. Both species of Giant Petrels are found in South Shetland, so this can be misleading. Table 1 is also a good example, in which one of the penguin spp. was written as: Gentoo penguin or Gentoo Penguin.

I cannot find the supplementary files. This should be corrected. Generally, the paper is not well-organized and not well-written and should be rewritten. I encourage authors to present some graphics recording to your data. For example, graph related to the number of samples from different localizations.

Author Response

Reviewer 4

The paper deals with an issue that is very timely, and it should be emphasized that any information that relates to the spread of avian influenza, in this very dangerous variant, is important and necessary, for decision-making, both environmental locally and globally. It should be borne in mind that bird flu in Antarctica can do incredible damage, and therefore the issue taken up by the authors is important.

However, I have a few comments and doubts, which concern both the form in which the authors wrote this paper and the formal/methodical side. Let me start with the latter.

First, all sources state that the highest risk of bird flu transmission to Antarctica, might be connected with migration of five species which nest in Antarctica regularly (i.e., kelp gull (Larus dominicanus), Antarctic tern (Sterna vittata), the southern giant petrel (Macronectes giganteus), the south polar skua (Stercorarius maccormicki)  and the brown skua (Stercorarius antarcticus)). Only 14 of 207 samples were taken from the above mentioned species. If this is the case, it cannot be concluded that there was no bird flu in Antarctica in early 2023. So the title of the paper is confusing and should be corrected accordingly to the work done.

Response: Partially accepted. We acknowledge that the most significant risk of avian influenza transmission to Antarctica is likely linked to the migratory patterns of five species that routinely nest on the continent. However, we also recognize that following the virus's introduction, it would be expected to impact other vulnerable species, such as penguins and elephant seals, which form large colonies and thus facilitate more straightforward evaluation and sample collection. This scenario mirrors occurrences in Peru, Argentina, and Chile in South America, where clinical observations of penguins and marine mammals were reported after the arrival of the virus by migratory birds. Then, the rationality of the sampling was incorporated.

Now 110-118.

The rationale behind the sampling strategy is related to the known susceptibility of penguins and elephant seals to HPAIV. We recognize that the primary risk of avian influenza reaching Antarctica most likely stems from migratory birds that regularly nest on the continent. We anticipate that upon the virus's introduction, it would likely affect other at-risk species such as penguins and elephant seals. These species, which form substantial colonies, allow for more efficient evaluation and sample collection. This pattern is reflected in events observed in Peru, Argentina, and Chile in South America, where the introduction of the virus by migratory birds led to documented clinical observations of HPAIV penguins and marine mammals [16–18]

Second, the choice of locations was not clearly described. Why exactly these locations were chosen. Besides, as far as Lions Rump is concerned, it is an ASPA, and sampling on it, requires the approval of the ethics committee. In the case of Chile, such approval is given by INACH, whether the team had such approval. If it did have it, why did it not take samples from the snowy sheathbill (Chionis albus) or brown skua (Stercorarius antarcticus), which nest there. It seems, that the sampling collection was done in haste and not necessarily in a thoughtful manner.  This needs clarifications. And the permit number for entrance to ASPA 151 should be stated clearly in the paper.

Response: Accepted. The rationale of the study about species was indicated in the previous comment. The rationale for the sample locations is improved in the text. These were locations where penguins and marine mammals were present, which aligns with the rationale of the species chosen. Plus, we attempt the collection of other flying birds such as those mentioned by the reviewer but compare them to penguins and elephant seals are more difficult to collect. The permit of entrance to ASPA 151 was incorporated (892/2022 INACH).

Third, the species names of animals should be standardized in accordance with accepted rules. This includes both English and Latin names. Sometimes authors wrote both words with a capital letter, sometimes whole guttural names, such as Sothern Giant Petrel, and sometimes short Giant Petrel. Both species of Giant Petrels are found in South Shetland, so this can be misleading. Table 1 is also a good example, in which one of the penguin spp. was written as: Gentoo penguin or Gentoo Penguin. Should be corrected accordingly.

Response: Accepted. All names were standardized.

I cannot find the supplementary files. This should be corrected. Generally, the paper is not well-organized and not well-written and should be rewritten. I encourage authors to present some graphics recording to your data. For example, graph related to the number of samples from different localizations.

Response: Accepted. The manuscript was revised, and sections were completely rewritten.

Comments on the Quality of English Language

Third, the species names of animals should be standardized in accordance with accepted rules. This includes both English and Latin names. Sometimes authors write both words with a capital letter, sometimes whole guttural names, such as Sothern Giant Petrel, and sometimes short Giant Petrel. Both species of Giant Petrels are found in South Shetland, so this can be misleading. Table 1 is also a good example, in which one of the penguin spp. was written as: Gentoo penguin or Gentoo Penguin.

Response: Accepted. All names were standardized.

I cannot find the supplementary files. This should be corrected. Generally, the paper is not well-organized and not well-written and should be rewritten. I encourage authors to present some graphics recording to your data. For example, graph related to the number of samples from different localizations.

Response: Accepted. The supplementary files were re-incorporated. The manuscript was improved in grammar and English. We consider that the table and the figure allow to understand the sampling.

Round 2

Reviewer 1 Report

Comments and Suggestions for Authors

The problems have been properly addressed and the revised conclusion is better supported by the results.

Reviewer 2 Report

Comments and Suggestions for Authors

All concerns have been addressed or further clarified. 

Reviewer 3 Report

Comments and Suggestions for Authors

Dear Authors,

I appreciate your efforts to make manuscript better. I also acknowledge difficulties working in the extreme environments, therefor limited number of samples in this study was not considered as disadvantage. Overall this study deserves to be published.

Reviewer 4 Report

Comments and Suggestions for Authors

This is the second time I have reviewed this work. I have to say that the work has been significantly changed and improved. I have received responses to all my comments, which have been taken into consideration in the new version of the paper. I have no, additional comments. The paper can be published.